# Trajectories of Depressive Individual Symptoms over Time during Transcranial Photobiomodulation

**Minoru Urata [1], Paolo Cassano [2,3], Richard Norton [4], Katelyn M. Sylvester [2], Koichiro Watanabe [1], Dan V. Iosifescu [5,6] and Hitoshi Sakurai [1,*]**

[1]  Department of Neuropsychiatry, Faculty of Medicine, Kyorin University, Tokyo 181-8611, Japan
[2]  Division of Neuropsychiatry and Neuromodulation, Department of Psychiatry, Massachusetts General Hospital, Boston, MA 02129, USA
[3]  Department of Psychiatry, Harvard Medical School, Boston, MA 02215, USA
[4]  Department of Psychological Sciences, University of Vermont, Burlington, VT 05405, USA
[5]  Clinical Research Division, Nathan Kline Institute for Psychiatric Research, Orangeburg, NY 10962, USA
[6]  Department of Psychiatry, New York University School of Medicine, New York, NY 10016, USA
*   Correspondence: hitoshi-sakurai@ks.kyorin-u.ac.jp; Tel.: +81-422-47-5511; Fax: +81-422-45-4697

**Abstract:** Transcranial photobiomodulation (t-PBM) is an innovative, non-invasive treatment for depression. This study aimed to investigate the changes in individual depressive symptoms during t-PBM treatment and identify the symptoms that improved in those who responded to treatment. The research analyzed data from two trials, the Evaluation of Light-emitting diodes Therapeutic Effect in Depression-2 and -3, focusing on patients with major depressive disorder. The patients received t-PBM treatment on the F3 and F4 regions of the scalp over eight weeks, with symptoms assessed weekly using the Quick Inventory for Depression Symptomatology (QIDS). A response was defined as a 50% or greater reduction in the QIDS score at eight weeks from baseline. Out of the 21 patients analyzed, 4 responded at eight weeks. Neurovegetative symptoms, including sleep disturbances and change in appetite, improved in ≥50% of the patients who had these symptoms at baseline. However, core depressive symptoms, including a depressed mood and lack of energy, persisted in about 80–90% of the patients. The responders showed a more than 75% improvement in these core depressive symptoms. These findings suggest that t-PBM treatment may uniquely alleviate certain neurovegetative symptoms in depression, and the improvement in core depressive symptoms might be linked to a clinical response to this treatment.

**Keywords:** depression; individual symptom; neurovegetative symptom; response; transcranial photobiomodulation (t-PBM)

## 1. Introduction

Antidepressant medications play a pivotal role in treating depression, with continuous treatment leading to remission in 89% of initially untreated patients within one year [1]. However, issues relating to tolerance often occur with these medications. A meta-analysis examining dropout rates from selective serotonin reuptake inhibitors and serotonin–norepinephrine reuptake inhibitors in 73 trials, involving 11,057 individuals, revealed a significantly higher risk ratio of discontinuation due to adverse events compared to the placebo [2]. Medication adherence also remains a challenge, with only approximately 50% adherence observed in outpatients with depression over a one-month period [3]. Moreover, given that psychotherapy, which includes cognitive-behavioral therapy, comes with time and cost constraints [4,5], and electroconvulsive therapy requires general anesthesia and raises concerns over cognitive impairment [6], there is a pressing need for a novel, safe, and effective treatment approach for depression.

Transcranial photobiomodulation (t-PBM) represents a recently developed alternative method for treating depression with near-infrared light. Animal studies have suggested that

tPBM yields local cerebral effects, encompassing neuroprotection, cognitive enhancement, neurogenesis, enhanced cerebral blood flow, and the regulation of neurotransmitters [7]. On a systemic level, it elicits a reduction in inflammation and improves mood and metabolic processes [7]. t-PBM's mechanism of action in humans is still debated; however, in this same Special Issue, it has been postulated that an increase in cerebral blood flow could be associated with an antidepressant effect [8]. Because t-PBM is device-based, it allows for easy self-administration at home with minimal training [9], thus making it more accessible compared to other neuromodulation treatments. Its effectiveness has been suggested in clinical studies for depression and anxiety, including one involving 10 depressed patients who demonstrated a mean decrease of 13.2 points on the 21-item Hamilton Depression Rating Scale (HAM-$D_{21}$) after two weeks of treatment [10]. Further, in a double-blind randomized controlled trial for 21 patients with depression, t-PBM showed a significantly greater reduction in the HAM-D score compared to the sham [11].

While the overall mean change is often utilized as the primary metric for treatment efficacy, individual depressive symptoms may vary in their responsiveness to different treatment approaches. For instance, an open-label study involving 811 depressed patients found that escitalopram was more effective in improving objective mood and cognitive symptoms, while nortriptyline was more effective for neurovegetative symptoms [12]. In a clinical data analysis in 180 depressed patients, repetitive transcranial magnetic stimulation (rTMS) treatment demonstrated limited improvement in neurovegetative symptom scores [13]. By clarifying which symptoms respond effectively to each approach for depression, we can tailor treatment to the individual needs of each patient. Furthermore, focusing on individual symptoms could assist in predicting treatment outcomes. Notably, early improvement in core depressive symptoms was reported to correlate with subsequent remission [14], while residual neurovegetative symptoms were associated with an increased risk of relapse [15].

To our knowledge, there have been no studies specifically examining the changes in individual depressive symptoms during t-PBM treatment. This study therefore aimed to investigate the trajectory of each depressive symptom among patients with depression and identify which symptoms show improvement among responders to t-PBM treatment.

## 2. Materials and Methods

### 2.1. Study Design

The data derived from the following two clinical trials were utilized in this report: the Evaluation of Light-emitting diodes Therapeutic Effect in Depression (ELATED)-2 trial [11] and the ELATED-3 trial [16]. Written informed consent was obtained from all participants after a comprehensive explanation of the study was provided. The Massachusetts General Hospital institutional review board approved both the ELATED-2 [11] and ELATED-3 trial [16], while the Nathan Kline Institute's institutional review board approved the ELATED-3 trial to be conducted as a second study site. Both ethical approvals for ELATED-2 and ELATED-3 were still in place at the time of these analyses, allowing for this secondary study on existing databases.

The ELATED-2 trial was an 8-week double-blind, sham-controlled trial that enrolled 21 patients aged 18–65 years diagnosed with major depressive disorder (MDD), according to the Diagnostic Statistical Manual, Fourth Edition (DSM-IV). These patients had a baseline HAM-$D_{17}$ total score that ranged from 14 to 24. The patients were assigned to receive either t-PBM with near-infrared (NIR) or sham treatment twice a week for 8 weeks, for a total of 16 sessions. The 16-item Quick Inventory for Depression Symptomatology, Self-Report (QIDS-$SR_{16}$) and HAM-$D_{17}$ were assessed at baseline and weekly thereafter.

The ELATED-3 trial was a 12-week double-blind, sham-controlled trial that employed a sequential parallel comparison design. It involved 54 patients aged between 18–70 years diagnosed with MDD, as per the DSM-IV. All participants had a baseline HAM-$D_{17}$ total score of 14 or above. The patients were randomized to receive t-PBM NIR or sham treatment in a 1:2 ratio for 6 weeks in phase 1. For phase 2, those who received t-PBM in phase 1

continued to receive t-PBM NIR for another 6 weeks, while non-responders to sham treatment in phase 1 were re-randomized to t-PBM NIR or sham treatment in a 1:1 ratio. t-PBM and sham treatment were administered twice a week in each phase. The patients were assessed with the HAM-D$_{17}$ at baseline, weeks 3, 6, 9, and 12, and with the 16-item Quick Inventory for Depression Symptomatology, Clinician-Rating (QIDS-C$_{16}$) at baseline and weekly thereafter.

### 2.2. Intervention

In the ELATED-2 trial, t-PBM (wavelength 823 nm, continuous wave, treatment area 28.7 cm$^2$, irradiance 36.2 mW/cm$^2$, maximum fluence 65.2 J/cm$^2$, 20–30 min/session) or sham was bilaterally administered to the dorsolateral prefrontal cortex twice a week for 8 weeks under double-blind randomized conditions. The LED t-PBM device utilized in the ELATED-2 trial was the Omnilux New U light emitting diode (manufactured by Photomedex Inc., Montgomeryville, PA, USA), with NIR or sham directed towards the F3 (left) and F4 (right) sites on the forehead (derived from the electroencephalography placement map), in order to target the dorsolateral prefrontal cortex. Concerning the ELATED-3 trial, t-PBM (wavelength 830 nm, continuous wave, treatment area 35.8 cm$^2$, irradiance 54.8 mW/cm$^2$, fluence 65.8 J/cm$^2$, and 20 min/session) or sham was bilaterally administered twice a week to the dorsolateral prefrontal cortex in a sequential parallel comparison design over 12 weeks. The ELATED-3 trial utilized the LED t-PBM device Transcranial PhotoBioModulation-1000 (TBPM-1000) manufactured by LiteCure LLV (New Castle, DE, USA) with NIR or sham directed at the F3 and F4 sites simultaneously with the Fp1 and Fp2 sites.

### 2.3. Statistical Analyses

Datasets were extracted for patients who underwent t-PBM or sham treatment for at least one week, had a baseline HAM-D$_{17}$ total score ranging between 14 and 24, and had weekly QIDS-SR$_{16}$ or QIDS-C$_{16}$ score data. To align with the 8-week duration of the ELATED-2 study, data from the first 8 weeks of the ELATED-3 study was included in our analysis (i.e., 6 weeks from phase 1 and the initial 2 weeks from phase 2). We analyzed patients who continuously received either t-PBM treatment or sham treatment (Figure 1). Patients who were re-randomized from sham to t-PBM treatment in the ELATED-3 study were excluded. Response was defined as a reduction of 50% or more in the QIDS-SR$_{16}$ or QIDS-C$_{16}$ score at 8 weeks from baseline. Baseline sociodemographic and clinical characteristics for t-PBM-treated patients were summarized separately for the responders and non-responders. Scores for each individual symptom on the QIDS-SR$_{16}$ or QIDS-C$_{16}$ were extracted at each week. Higher scores were noted for the composite questions (i.e., decrease and increase) regarding appetite and weight. Proportions of patients with each symptom at endpoint, among those who had the symptom at the baseline, were identified using last observation carried forward (LOCF) analysis. Average trajectories of mean scores over time in each individual symptom for both responders and non-responders to t-PBM treatment were also estimated using available case (AC) and LOCF analyses.

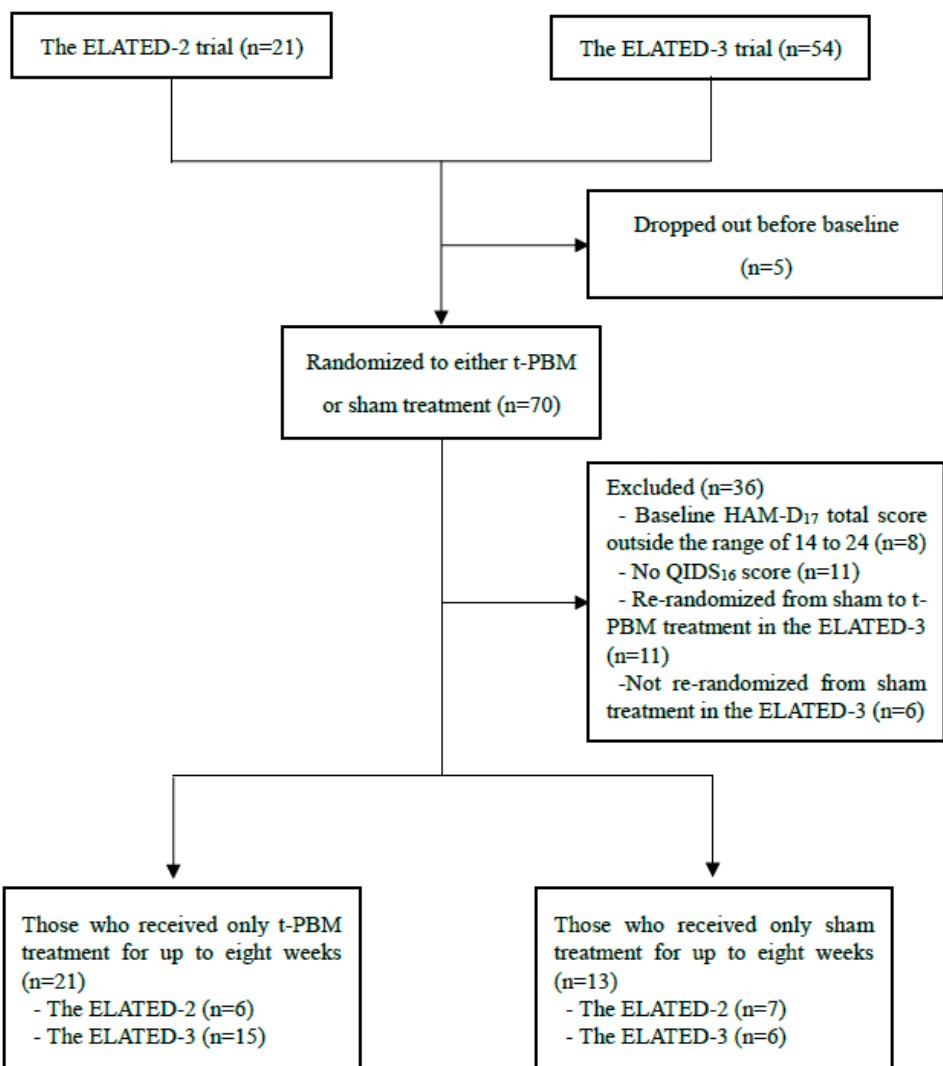

**Figure 1.** Patient flow.

## 3. Results

### 3.1. Patient Disposition and Characteristics

Of the 21 patients enrolled for t-PBM treatment, 4 responded at 8 weeks while the other 17 did not. Table 1 outlines baseline demographic and clinical characteristics of these patients. Similarly, of the 13 patients who underwent sham treatment, 3 responded at 8 weeks with the other 10 showing no response.

### 3.2. Persistent Baseline Symptoms

In the t-PBM-treated patients displaying significant symptoms at baseline (i.e., a score of one or higher on the QIDS-SR$_{16}$ or QIDS-C$_{16}$), certain symptoms were completely improved in over half the patients after 8 weeks of the t-PBM treatment, as per the LOCF analysis (Table 2). These symptoms include hypersomnia (with a persistence rate of 30.0%, meaning that, out of the 10 patients who had the symptom at baseline, 3 still exhibited the symptom after 8 weeks of treatment), slowing down (30.0%, 3 out of 10), early morning insomnia (33.3%, 3 out of 9), and appetite change (46.2%, 6 out of 13). Conversely, more than 75% of the patients who initially exhibited the following symptoms reported the persistence of symptoms after eight weeks of t-PBM treatment: sad mood (90.5%, 19 out of 21), mid-nocturnal insomnia (86.7%, 13 out of 15), lack of energy (85.7%, 18 out of 21), reduced involvement (85.0%, 17 out of 20), and suicidal ideation (75.0%, 9 out of 12).

**Table 1.** Baseline demographic and clinical characteristics of t-PBM-treated patients.

| Characteristics | Responders (*n* = 4) | Non-Responders (*n* = 17) |
|---|---|---|
| Age in years, mean $\pm$ SD | 47.3 $\pm$ 16.5 | 38.4 $\pm$ 15.5 |
| Gender, female, *n* (%) | 4 (100.0%) | 13 (76.5%) |
| Race, *n* (%) | | |
|     White | 3 (75.0%) | 14 (82.4%) |
|     Asian | 1 (25.0%) | 2 (11.8%) |
|     Black | 0 (0.0%) | 1 (5.9%) |
| Ethnicity, *n* (%) | | |
|     Hispanic or Latino | 0 (0.0%) | 1 (5.9%) |
|     Not Hispanic or Latino | 3 (75.0%) | 14 (82.4%) |
|     Not specified | 1 (25.0%) | 2 (11.8%) |
| HAM-D$_{17}$, mean $\pm$ SD | 21.5 $\pm$ 3.3 | 19.9 $\pm$ 2.9 |
| QIDS, mean $\pm$ SD | 14.0 $\pm$ 2.1 | 14.9 $\pm$ 2.7 |

HAM-D$_{17}$, 17-item Hamilton Depression Rating Scale; QIDS, Quick Inventory for Depression Symptomatology; SD, standard deviation; t-PBM, transcranial photobiomodulation.

**Table 2.** Proportion of persistent baseline symptoms.

| | Symptom at Baseline | Symptom at Endpoint among Those Who Had It at Baseline |
|---|---|---|
| Sleep onset insomnia | 71.4% (15/21) | 73.3% (11/15) |
| Mid-nocturnal insomnia | 71.4% (15/21) | 86.7% (13/15) |
| Early morning insomnia | 42.9% (9/21) | 33.3% (3/9) |
| Hypersomnia | 47.6% (10/21) | 30.0% (3/10) |
| Sad mood | 100.0% (21/21) | 90.5% (19/21) |
| Appetite change | 61.9% (13/21) | 46.2% (6/13) |
| Weight change | 66.7% (14/21) | 57.1% (8/14) |
| Concentration/decision making | 90.5% (19/21) | 73.7% (14/19) |
| Negative self-view | 85.7% (18/21) | 72.2% (13/18) |
| Suicidal ideation | 57.1% (12/21) | 75.0% (9/12) |
| Reduced involvement | 95.2% (20/21) | 85.0% (17/20) |
| Lack of energy | 100.0% (21/21) | 85.7% (18/21) |
| Slowing down | 47.6% (10/21) | 30.0% (3/10) |
| Restlessness | 52.4% (11/21) | 72.7% (8/11) |

For sham-treated patients with significant symptoms at baseline, more than half reported a complete improvement after 8 weeks for symptoms like restlessness (33.3%, 2 out of 6) and suicidal ideation (44.4%, 4 out of 9) (data not shown). Yet, over 75% of the patients initially showing the following symptoms still reported them after eight weeks of sham treatment: weight change (87.5%, 7 out of 8), sleep-onset insomnia (85.7%, 6 out of 7), sad mood (84.6%, 11 out of 13), concentration loss (83.3%, 10 out of 12), lack of energy (81.8%, 9 out of 11), early morning insomnia (80.0%, 4 out of 5), mid-nocturnal insomnia (76.9%, 10 out of 13), and reduced involvement (76.9%, 10 out of 13).

*3.3. Trajectories of Individual Symptoms over Time during t-PBM*

In the patients who responded to t-PBM treatment, there was a 50% or greater improvement observed in the QIDS-SR$_{16}$ or QIDS-C$_{16}$ scores for all symptoms besides sleep—onset insomnia, mid-nocturnal insomnia, and early morning insomnia after 8 weeks of treatment as compared to the baseline in the LOCF analysis (Supplementary Figure S1). Moreover, the responders exhibited improvement rates of over 75% for symptoms including sad mood, appetite change, difficulties with concentration and decision making, lack of energy, slowing down, and restlessness. Figure 2 illustrates the trajectories of some typical symptoms over time. In contrast, among the patients who did not respond, only hypersomnia showed a 50% or greater improvement.

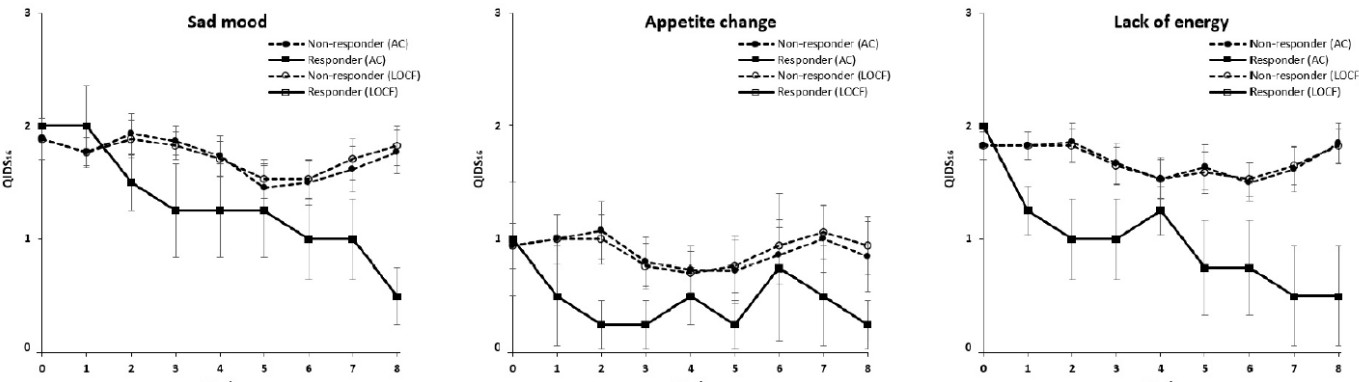

**Figure 2.** Trajectories of individual symptoms over time in responders and non-responders to t-PBM treatment (three typical symptoms). AC, available case; LOCF, last observation carried forward; t-PBM, transcranial photobiomodulation.

## 4. Discussion

While this study was conducted with a relatively small sample size, which significantly limits the generalizability of the results, it represents the first investigation into the changes in individual symptoms among depressed patients undergoing t-PBM treatment. Over the 8-week t-PBM treatment, a high percentage of patients initially presenting with neurovegetative symptoms (i.e., early morning insomnia, hypersomnia, and appetite change) experienced an improvement in these symptoms at the endpoint, which was not observed in the sham treatment. Core depressive symptoms including depressed mood and lack of energy persisted in many patients receiving t-PBM treatment. On the other hand, the responders to the t-PBM treatment demonstrated improvements in the average scores for these core depressive symptoms. These findings suggest that t-PBM treatment may have a unique efficacy in alleviating neurovegetative symptoms in all patients, while an improvement in core symptoms could be associated with treatment response.

Regarding neurovegetative symptoms (including early morning awakening, hypersomnia, and appetite change), over half of the patients with these symptoms at baseline experienced the resolution of these symptoms following eight weeks of t-PBM treatment. This contrasts with a past study which explored the trajectory of individual symptoms in 180 depressed patients who received rTMS treatment and showed minimal improvement in average scores for neurovegetative symptoms based on the QIDS-SR$_{16}$ [13]. Similarly, a study re-analyzing the individual symptoms of 2874 patients with depression receiving citalopram in the Sequenced Treatment Alternatives to Relieve Depression (STAR*D) trial found a poor improvement in oversleeping and weight changes in non-remitters [14]. The contrast with these findings indicates a specific efficacy of t-PBM treatment in symptom improvement, distinct from that of pharmacotherapy or rTMS treatment.

The similarities between the effects of t-PBM treatment on neurovegetative symptoms and another non-invasive approach, bright light therapy, are worth noting. Bright light therapy, which involves the daily exposure of the entire body to light exceeding 2500 lux from fluorescent lamps, primarily works through the visual system [17]. According to a comprehensive review of 53 studies, this therapy significantly improved sleep disturbances in patients suffering from various sleep disorders [18]. Furthermore, a randomized clinical trial with 70 patients with Alzheimer's disease found noteworthy changes in appetite, as measured by the Neuropsychiatric Inventory-Nursing Home version scale, after 10 weeks of daily high-intensity light therapy with an intensity exceeding 2500 lux [19]. However, despite the comparable outcomes, the mechanisms through which these therapies work appear to be distinct. Bright light therapy is believed to affect melatonin secretion [20], while t-PBM seems to predominantly influence mitochondrial activity, as animal studies have suggested [21–25]. It is worth noting that mitochondrial dysfunction may potentially contribute to weight loss within the domain of cancer cachexia [26]. Additionally, animal

studies have demonstrated the association between abnormalities in mitochondrial function and weight change [27,28]. The role of mitochondria in regulating sleep mechanisms is also of considerable interest [29,30]. For instance, studies in animals have shown that mitochondrial function may be related to neurological alterations caused by sleep deprivation in rats [31]. The improvement in neurovegetative symptoms observed in this study, such as appetite and weight changes, along with sleep disturbances, following t-PBM treatment may be linked to its influence on mitochondrial activity. Consequently, it is essential to delve deeper into research to understand why these two different non-invasive therapies, t-PBM treatment and bright light therapy, result in similar therapeutic benefits.

In the present study, the scores for core depressive symptoms in the QIDS, including depressed mood and lack of energy, began to improve from the second week in the responders to the t-PBM treatment, while the non-responders showed minimal improvement. Considering core depressive symptoms display a heightened correlation with other symptoms [32], their improvement may lead subsequently to overall symptom improvement. A re-analysis of the STAR*D trial reported that an early improvement in core depressive symptoms within two weeks predicted a higher probability of eventual remission [14]. Another study involving 67 patients treated with duloxetine found that early improvement in specific individual symptoms, including dysphoria and retardation, after four weeks of intervention significantly correlated with subsequent remission [33]. Assessing individual symptoms in conjunction with the overall depression assessment score could enhance our ability to predict treatment outcomes and allow for more personalized therapeutic strategies [34–36].

This study has several limitations. First, the dataset utilized in the present analysis combines the results from the ELATED-2 and ELATED-3 trials. Neither of these studies were explicitly designed to evaluate each individual depressive symptom. Second, the sample size was small, particularly with only four individuals in the responders to the t-PBM treatment, which hampers generalizability. Third, the study population was predominantly composed of women (80%) and Caucasians (80%), making it necessary to apply the results cautiously to other demographics. Finally, the concomitant use of pharmacotherapy or psychotherapy by some participants could influence the study results.

In conclusion, t-PBM treatment appears to offer potential for improving neurovegetative symptoms including sleep disturbances and appetite change. It also demonstrated improvements in core depressive symptoms among those who responded to the treatment. Further analysis of individual symptoms utilizing larger sample sizes will be pivotal in fully appreciating the efficacy and outcomes of t-PBM treatment.

**Supplementary Materials:** The following supporting information can be downloaded at https://www.mdpi.com/article/10.3390/photonics10121324/s1, Figure S1: Trajectories of individual symptoms over time during tPBM in responders and non-responders (all symptoms). AC, available case; LOCF, last observation carried forward; t-PBM, transcranial photobiomodulation.

**Author Contributions:** Conceptualization, H.S.; methodology, H.S.; investigation, M.U.; writing—original draft preparation, M.U.; writing—review and editing, R.N., K.M.S., K.W., D.V.I. and H.S.; visualization, M.U.; project administration, P.C., D.V.I. and H.S.; funding acquisition, P.C. and D.V.I. All authors have read and agreed to the published version of the manuscript.

**Funding:** The ELATED-2 trial was funded by the Harvard Psychiatry Department (Dupont-Warren Fellowship and Livingston Award) and the Brain and Behavior Research Foundation (NARSAD Young Investigator Award; Grant No. 19159). Devices and technical support for the ELATED-2 Trial were provided by Photomedex, Inc., Montgomeryville, PA. The ELATED-3 trial was funded by Litecure LLC, 101 Lukens Dr, New Castle, DE 19720. The sponsor also provided the study devices and related technical support during the trial.

**Institutional Review Board Statement:** Given the completely anonymous nature of the post hoc analysis and an absence of any direct human involvement, ethical approval was not sought.

**Informed Consent Statement:** Written informed consent was obtained from all participants after a comprehensive explanation of the study was provided.

**Data Availability Statement:** Data is contained within the article and Supplementary Materials.

**Acknowledgments:** This study was supported by the Japan Society for the Promotion of Science KAKENHI under grant numbers JP22K15755 (H.S.), Japan Research Foundation for Clinical Pharmacology (H.S.), and Takeda Science Foundation (H.S.).

**Conflicts of Interest:** Urata has received speaker honoraria from Sumitomo Pharma and Janssen Pharmaceutical over the last three years. Cassano consulted for Janssen Research and Development and for Niraxx Light Therapeutics Inc. Cassano was funded by PhotoThera Inc. (CA, US), LiteCure LLC. (TX, US) and Cerebral Sciences Inc. (CA, US) to conduct studies on transcranial photobiomodulation. Cassano is a share-holder of Niraxx Light Therapeutics Inc. (CA, US). Cassano has filed several patents related to the use of nearinfrared light in psychiatry. Norton and Sylvester have no financial interests to declare. Watanabe has received manuscript fees or speaker's honoraria from Eisai, Eli Lilly, Janssen Pharmaceutical, Kyowa Pharmaceutical, Lundbeck Japan, Meiji Seika Pharma, Mitsubishi Tanabe Pharma, MSD, Otsuka Pharmaceutical, Pfizer, Shionogi, Sumitomo Pharma, and Takeda Pharmaceutical and received research/grant support from Daiichi Sankyo, Eisai, Meiji Seika Pharma, Mitsubishi Tanabe Pharma, MSD, Otsuka Pharmaceutical, Pfizer, Sumitomo Pharma, and Takeda Pharmaceutical. In addition, Watanabe is a consultant of Boehringer Ingelheim, Daiichi Sankyo, Eisai, Eli Lilly, Janssen Pharmaceutical, Kyowa Pharmaceutical, Lundbeck Japan, Luye Pharma, Mitsubishi Tanabe Pharma, Otsuka Pharmaceutical, Pfizer, Sumitomo Dainippon Pharma, Taisho Toyama Pharmaceutical, and Takeda Pharmaceutical. In the last 10 years, Iosifescu has served as a consultant for Alkermes, Allergan, Angelini, Axsome, Biogen, Boehringer Ingelheim, the Centers for Psychiatric Excellence, Clexio, Jazz, Lundbeck, Neumora, Otsuka, Precision Neuroscience, Relmada, Sage Therapeutics, and Sunovion. He has received grant support (paid to his institutions) from Alkermes, AstraZeneca, BrainsWay, LiteCure, NeoSync, Otsuka, Roche, and Shire. Sakurai has received manuscript fees and speaker honoraria from Eisai, Kyowa Pharmaceutical Industry, Lundbeck, Meiji Seika Pharma, Otsuka Pharmaceutical, Shionogi Pharma, Sumitomo Pharma, Takeda Pharma, MSD, and Yoshitomi Yakuhin over the last three years.

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
