# Peer review of "Trajectories of Depressive Individual Symptoms over Time during Transcranial Photobiomodulation"

_photonics, doi:10.3390/photonics10121324_

Round 1

Reviewer 1 Report

Comments and Suggestions for Authors

I think this secondary analysis of ELATED-2 and ELATED-3 is relevant and well written. 

There is one thing worth considering. Since it seems that PBM was associated with notable improvement of neurovegetative symptoms, it would be nice to have some comparative data about the participants receiving placebo treatment, assuming that the participants receiving placebo probably also saw some improvements as well.

Author Response

Comments 1: I think this secondary analysis of ELATED-2 and ELATED-3 is relevant and well written. There is one thing worth considering. Since it seems that PBM was associated with notable improvement of neurovegetative symptoms, it would be nice to have some comparative data about the participants receiving placebo treatment, assuming that the participants receiving placebo probably also saw some improvements as well.

Response 1: Thank you for your valuable suggestion. In line with your recommendation, we have now examined the proportions of patients who displayed each individual symptom at the endpoint after having presented these symptoms at baseline and undergoing sham treatment for eight weeks. The improvement in neurovegetative symptoms noted in patients treated with t-PBM was not commonly observed in those who received sham treatment.

Changes to Manuscript

(P8 L8 – P8 L15) "Datasets were extracted for patients who underwent t-PBM or sham treatment for at least one week, had a baseline HAM-D17 total score ranging between 14 and 24, and had weekly QIDS-SR16 or QIDS-C16 score data. To align with the 8-week duration of the ELATED-2 study, data from the first 8 weeks of the ELATED-3 study was included in our analysis (i.e. 6 weeks from phase 1 and the initial 2 weeks from phase 2). We analyzed patients who continuously received either t-PBM treatment or sham treatment (Figure 1). Patients who were re-randomized from sham to t-PBM treatment were excluded."

(P10 L6 – P10 L7) "Similarly, of the 13 patients who underwent sham treatment, 3 responded at 8 weeks with the other 10 showing no response."

(P11 L5 – P11 L12) "For sham-treated patients with significant symptoms at baseline, more than half reported complete improvement after 8 weeks for symptoms like restlessness (33.3%, 2 out of 6) and suicidal ideation (44.4%, 4 out of 9) (data not shown). Yet, over 75% of the patients initially showing the following symptoms still reported them after eight weeks of sham treatment: weight change (87.5%, 7 out of 8), sleep onset insomnia (85.7%, 6 out of 7), sad mood (84.6%, 11 out of 13), concentration loss (83.3%, 10 out of 12), lack of energy (81.8%, 9 out of 11), early morning insomnia (80.0%, 4 out of 5), mid-nocturnal insomnia (76.9%, 10 out of 13), and reduced involvement (76.9%, 10 out of 13)."

(P13 L4 – P13 L8) "Over the 8-week t-PBM treatment, a high percentage of patients initially presenting with neurovegetative symptoms (i.e., early morning insomnia, hypersomnia, and appetite change) experienced the improvement of these symptoms at the endpoint, which was not observed in sham treatment."

Reviewer 2 Report

Comments and Suggestions for Authors

The scientific paper "Trajectories of depressive individual symptoms over time during transcranial photobiomodulation" aimed to investigate the trajectory of each depressive symptom among patients with depression and identify which symptoms show improvement in responders to t-PBM treatment. After carefully reading the manuscript, I can suggest that you make the following changes:

1)      Authors must be aware of the rules of the Photonics MDPI journal. Note on the electronic page in the instructions to authors that, for example, the titles of the authors (Phd, etc.) are not included, they must not have an Abbreviated Title, among others.

2)      The number of research participants (21) is too low to support the results and conclusions of a clinical study.

3)      The introduction is very poor about photobiomodulation. it should incorporate the beneficial effects in general first, its main actions of biomodulating the body's response after its application.

4)      In the methodology, in the study design, it was not clear the distribution in groups of the participants, how they were randomized and selection criteria (inclusion and exclusion). I suggest a graphic figure typical of clinical studies.

5)      What equipment was used for the PBM? Insert manufacturer, city and country). What is the area of the beam?

6)      Figure 1 has 14 graphs and is impossible to analyze. I suggest distributing it in 2 to 3 figures, with larger and more visible graphics.

7)      Ethical approval is extremely necessary. I do not consider the justification presented by the authors to be valid.

8)      Only 22 references to a clinical study already demonstrate that it was carried out without scientific basis.

Comments on the Quality of English Language

Moderate editing of English language required

Author Response

Comments 1: Authors must be aware of the rules of the Photonics MDPI journal. Note on the electronic page in the instructions to authors that, for example, the titles of the authors (Phd, etc.) are not included, they must not have an Abbreviated Title, among others.

Response 1: Thank you for pointing this out. We have modified the manuscript according to the instructions for authors.

Comments 2: The number of research participants (21) is too low to support the results and conclusions of a clinical study.

Response 2: We are grateful for your feedback, which is totally agreeable. We have addressed this limitation in the discussion section, emphasizing that the results should be interpreted with caution.

Changes to Manuscript

(P13 L2 – P13 L4) “While this study was conducted with a relatively small sample size, which significantly limits the generalizability of the results, it represents the first investigation into the changes in individual symptoms among depressed patients undergoing t-PBM treatment.”

(P15 L12 – P15 L14) “Second, the sample size was small, particularly with only four individuals in the responders to t-PBM treatment, which hampers generalizability."

Comments 3: The introduction is very poor about photobiomodulation. it should incorporate the beneficial effects in general first, its main actions of biomodulating the body's response after its application.

Response 3: Thank you very much for sharing your constructive feedback. We understand your point, at the same time we have written many of these introductions on t-PBM and are very familiar about the literature. The key aspect is that there is very little knowledge over what in fact are the mechanisms of actions that lead to an effect in the case of t-PBM. The whole bioenergetic effects of t-PBM on neuronal cyt-C oxidase have been considered improbable by many, based on poor penetration. We believe that a more conservative and agnostic approach in this paper is more appropriate given that the scope of the paper is more clinical. Other papers which are more methodological have already extensively covered this mechanistic component of t-PBM within this same special issue. We have mentioned the effects on cerebral blood flow as a potential mechanism in the introduction.

Changes to Manuscript

(P4 L15 – P4 L18) “Transcranial photobiomodulation (t-PBM) represents a recently developed alternative method for treating depression with near-infrared light. t-PBM mechanism of action in humans is still debated, however, in this same special issue, it has been postulated that an increase on cerebral blood flow could be associated with antidepressant effect [7]."

Comments 4: In the methodology, in the study design, it was not clear the distribution in groups of the participants, how they were randomized and selection criteria (inclusion and exclusion). I suggest a graphic figure typical of clinical studies.

Response 4: We really appreciate your constructive suggestion. We have provided visual representation of the patients examined in this study, which you can find as Figure 1.

Changes to Manuscript

Figure 1 has been added.

Comments 5: What equipment was used for the PBM? Insert manufacturer, city and country). What is the area of the beam?

Response 5: We have included the majority of the information you requested. However, the manufacturer has not supplied the beam area measurements for these devices, as they are modified versions of devices that are already on the market.

Changes to Manuscript

(P7 L17 – P7 L21) “The LED t-PBM device utilized in the ELATED-2 trial was the Omnilux New U light emitting diode (manufactured by Photomedex Inc., Montgomeryville PA), with NIR or sham directed towards the F3 (left) and F4 (right) sites on the forehead (derived from the electroencephalography placement map) in order to target the dorsolateral prefrontal cortex."

(P8 L2 – P8 L5) “The ELATED-3 trial utilized the LED t-PBM device Transcranial PhotoBioModulation-1000 (TBPM-1000) manufactured by LiteCure LLV (New Castle, DE) with NIR or sham directed at the F3 and F4 sites simultaneously with the Fp1 and Fp2 sites."

Comments 6: Figure 1 has 14 graphs and is impossible to analyze. I suggest distributing it in 2 to 3 figures, with larger and more visible graphics.

Response 6: Thank you for your insightful suggestion. We concur with your observation and have accordingly retained the three key graphs as main figures in the paper. The earlier versions of these figures, which include all individual symptoms, have been appropriately moved to the Supplementary Materials section.

Changes to Manuscript

Figure 2 has been modified.

Supplementary Figure 1 has been added.

Comments 7: Ethical approval is extremely necessary. I do not consider the justification presented by the authors to be valid.

Response 7: We have clarified this point in the revised version. Thank you very much.

Changes to Manuscript

(P6 L10 – P6 L12) “Both ethical approvals for ELATED-2 and ELATED-3 were still in place at the time of these analyses allowing for this secondary study on existing databases.”

Comments 8: Only 22 references to a clinical study already demonstrate that it was carried out without scientific basis.

Response 8: Thank you for your comment. We have supplemented our study with additional references where relevant. The aim of this secondary study is to delve deeper into the clinical trajectories of symptoms, rather than to present a comprehensive review of the existing literature on t-PBM. For a more thorough literature perspective, we invite readers to consult other papers from our group featured in this special issue, which offer an expansive examination of t-PBM in the context of major depressive disorder.

Changes to Manuscript

References 7 and 21-24 have been added.

4. Response to Comments on the Quality of English Language

Point 1: Moderate editing of English language required

Response 1:     An expert in English has thoroughly reviewed and meticulously edited the manuscript for language clarity. We appreciate your attention to detail.

Reviewer 3 Report

Comments and Suggestions for Authors

Photobiomodulation lead to 75% improvement in neurovegetative symptoms at least in responders:

Abstract: What do mean with "MIGHT be linked to a clinical response ..."

Discussion: You write "...it is essential to delve deeper into research..." besides the reference 19 there are so many other references, which show a causal link between t-PBM or at least infrared impingement and e.g. positive modulation of the respiratory chain etc. Please cite more of these findings.

Minor point: in table 2 you write "energy" and "involvement" better write "lack of energy" and "reduced involvement" like in the text.  

Author Response

Comments 1: Abstract: What do mean with "MIGHT be linked to a clinical response ..."

Response 1: After an 8-week t-PBM treatment, responders demonstrated considerable improvements in multiple core depressive symptoms. This indicates a potential correlation between the improvement of these core symptoms and the clinical response to the treatment. Thank you very much.

Comments 2: Discussion: You write "...it is essential to delve deeper into research..." besides the reference 19 there are so many other references, which show a causal link between t-PBM or at least infrared impingement and e.g. positive modulation of the respiratory chain etc. Please cite more of these findings.

Response 2: We are grateful for your valuable input. We have followed your advice and cited more papers supporting the classical metabolic theory over the neuromodulatory effects of t-PBM.

Changes to Manuscript

References 7 and 21-24 have been added.

Comments 3: Minor point: in table 2 you write "energy" and "involvement" better write "lack of energy" and "reduced involvement" like in the text. 

Response 3: Thank you for pointing this out, which is totally agreeable. We have implemented the revisions in accordance with your feedback.

Round 2

Reviewer 2 Report

Comments and Suggestions for Authors

Dear Authors

The figures were not inserted in the body of the manuscript. This way the evaluation for review is compromised. Please be aware that figures and tables must be in the body of the manuscript. In fact, figures 1 and 2 are not even in the supplementary material. I think the introduction could be even better, describing more about photobiomodulation and its local and systemic effects, regardless of whether the reader is knowledgeable on the subject, as some will not be. I also consider the number of references low, to support the discussion.

Comments on the Quality of English Language

Minor editing

Author Response

Comment 1: The figures were not inserted in the body of the manuscript. This way the evaluation for review is compromised. Please be aware that figures and tables must be in the body of the manuscript. In fact, figures 1 and 2 are not even in the supplementary material.

Response 1: The tables and figures have now been included in the body of the manuscript. Additionally, figures have been added to the supplementary material.

Comment 2: I think the introduction could be even better, describing more about photobiomodulation and its local and systemic effects, regardless of whether the reader is knowledgeable on the subject, as some will not be.

Response 2: Thank you for your valuable suggestion. We added to the introduction the local and systemic effects of t-PBM treatment.

Changes to Manuscript

(P4 L16 – P4 L20) “Animal studies have suggested that tPBM yields local cerebral effects, encompassing neuroprotection, cognitive enhancement, neurogenesis, enhanced cerebral blood flow, and regulation of neurotransmitters [7]. On a systemic level, it elicits a reduction in inflammation and improves mood and metabolic processes [7]."

Comment 3: I also consider the number of references low, to support the discussion.

Response 3: Thank you for your comment. In order to enrich the discussion, several references have been added to the text.

Changes to Manuscript

(P16 L16 – P17 L4) “It is worth noting that mitochondrial dysfunction may potentially contribute to weight loss within the domain of cancer cachexia [26]. Additionally, animal studies have demonstrated the association between abnormalities in mitochondrial function and weight change [27,28]. The role of mitochondria in regulating sleep mechanisms is also of considerable interest [29,30]. For instance, studies in animals have shown that mitochondrial function may be related to neurological alterations caused by sleep deprivation in rats [31]. The improvement in neurovegetative symptoms observed in this study, such as appetite and weight changes, along with sleep disturbances, following t-PBM treatment, may be linked to its influence on mitochondrial activity. Consequently, it is essential to delve deeper into research to understand why these two different non-invasive therapies, t-PBM treatment and bright light therapy, result in similar therapeutic benefits."

(P17 L16 – P17 L18) “Assessing individual symptoms in conjunction with the overall depression assessment score could enhance our ability to predict treatment outcomes and allow for more personalized therapeutic strategies [34,35,36]."
